# Sensitivity-Analysis-Driven Surrogate Model for Molten Salt Reactors Control

Eric Cervi [1], Xuefei Lu [2], Antonio Cammi [1,*], Francesco Di Maio [1] and Enrico Zio [1,3,4]

1   Department of Energy, Politecnico di Milano, Via La Masa 34, 20156 Milano, Italy
2   SKEMA Business School, Université Côte d'Azur, 5 Quai Marcell Dassault, 92150 Paris, France
3   Mines Paris, PSL Research University, CRC, 06904 Sophia Antipolis, Paris, France
4   Department of Nuclear Engineering, College of Engineering, Kyung Hee University, Seoul 02453, Korea
*   Correspondence: antonio.cammi@polimi.it

**Abstract:** The numerical analysis for the controllability assessment of a new design nuclear reactor is typically carried out by means of complex multiphysics codes, solving high fidelity partial differential equations governing the system neutronics as well as the fluid dynamics. Multiphysics codes deliver very accurate solutions at the expense of high computational times, which could be of several hours depending on the specific case study. In this work, to efficiently reduce runtimes, a sensitivity analysis (SA) is carried out to identify the most important input parameters affecting the solution of a multiphysics model developed for the controllability assessment of molten salt reactors (MSRs). The numerical modeling of these innovative systems is fundamental to allow for a safer and more sustainable power production (e.g., due to the lower radiotoxicity of the actinide inventory in MSRs and to the possibility of operation at atmospheric pressure). In this paper, four global sensitivity measures are calculated first, including the Pearson correlation coefficient, δ, Kolmogorov–Smirnov and Kuiper indices, whose results are aggregated by an ensemble strategy and confirmed by the CUmulative SUm of NOrmalized Reordered Output (CUSUNORO) plot. The results of the SA point out that the fuel density is the most important parameter yielding the largest variations in the system reactivity, fundamental for guaranteeing the MSR controllability. In light of this result, a simplified, surrogate model is then developed, which uses density as the only input parameter to determine reactivity, guaranteeing runtime reductions from several hours to a few seconds and, at the same time, a comparable level of accuracy of the multiphysics model. This result demonstrates the capability of global sensitivity analysis approaches to effectively identify the most relevant parameters in MSR systems, supporting the development of simplified, control-oriented models for these innovative reactors.

**Keywords:** molten salt reactor (MSR); multiphysics; OpenFOAM; sensitivity analysis

## 1. Introduction

Molten salt reactors (MSRs) are circulating fuel nuclear reactors in which a mixture of molten thorium and uranium fluorides acts as fuel and coolant simultaneously [1]. In recent years, MSRs are gathering a strong interest from the nuclear research community, due to their intrinsic characteristics of safety and sustainability. Thanks to the high boiling point of the molten salts, MSRs can be operated at atmospheric pressure. In addition, the adoption of a closed thorium fuel cycle may lead to an actinide inventory with lower radiotoxicity. Finally, the adoption of a liquid fuel can potentially allow for a significant plant simplification (due to the core homogeneity), as well as a greater compactness as the fission energy is directly released into the coolant [2]. For these reasons, MSRs are strong candidates for a cleaner, safer, and more sustainable power production. At the same time, MSRs raise new technical challenges that call for new simulation tools, tailored to the specificities of these innovative systems. In more detail, compared to traditional solid-fueled reactors, the circulating fuel is a distinguishing feature of MSRs [3,4], leading

to completely new design and related technological challenges. Notably, the delayed neutron precursors are not spatially static, as in conventional nuclear systems, but they are dragged by the fuel mixture through the reactor and the external circuits. For this reason, delayed neutrons can be emitted in peripheral regions of the reactor, where the neutron importance with respect to reactivity is lower, or even in the external circuit, where they do not contribute at all to fissions. As a consequence, the coupling between the neutronics and thermo-fluid dynamics of the system is even stronger than in traditional reactors, since the fuel velocity field directly influences the precursor distribution.

Multiphysics approaches provide a good way to inherently couple all the physical phenomena occurring in the reactor in the same simulation environment, offering an efficient way to handle the coupling non-linearities [5]. Multiphysics models of the MSRs have been developed (Cammi et al. [6,7], Aufiero et al. [8,9], Fiorina et al. [10,11]), coupling the neutron diffusion equation with a single-phase, incompressible thermal-hydraulics model, where transport equations for the moving precursors have also been implemented [9]. Furthermore, a Finite Element multiphysics model for MSRs was developed in [12–14], while de Oliveira et al. proposed a Finite Volume model using the GeN-Foam code [15]. In addition, capabilities to simulate fuel salt solidification and draining transients in MSRs were developed by Tano et al. [16,17].

In some MSR designs [1], the injection of helium bubbles is foreseen for the removal of gaseous fission products. These helium bubbles not only influence reactivity, due to their negative void coefficient, but also increase the liquid fuel compressibility, which is expected to have a strong impact on fast, reactivity-driven transients, where the finite propagation velocity of pressure waves can lead to delays in the expansion reactivity feedback [18]. Therefore, careful investigation is needed in order to assess the safety of MSRs, also considering the presence of gas bubbles inside the reactor. Both the bubble motion and compressibility effects cannot be described by means of standard single-phase, incompressible thermal-hydraulics models. To address these issues, a multiphysics OpenFOAM solver has been developed in [19,20], coupling the multi-group neutron diffusion equation with a two-fluid, or Euler–Euler [21] model for two-phase, compressible thermal-hydraulics, where the motion of precursors in the liquid fuel is also considered.

These complex multiphysics tools can produce very accurate solutions, being able to catch phenomena that could not be described with simpler approaches. However, this accuracy comes at the expense of large computational times, on the order of several hours.

In order to reduce the computational time for a real-time control analysis and decision support, a surrogate model of the multiphysics model can be beneficial. In this work, sensitivity analysis (SA) methods are first used to identify the most important input parameters of the multiphysics model, to be used to build a simplified surrogate model of the MSR behavior. In general, SA methods fall into two categories: local or global. Local sensitivity approaches like the Tornado diagrams [22] focus on the simulation behavior around certain reference values of the inputs; global sensitivity methods such as the variance-based methods [23] allow the inputs to vary in their entire input [24]. It has been shown that the ensemble of different SA methods can obtain robust and reliable results [25]. In this work, we ensemble four commonly used global sensitivity methods: Pearson correlation coefficient, the Borgonovo δ index, the Kolmogorov–Smirnov index, and the Kuiper index [26]. The uncertainty in the estimates is quantified via bootstrapping and the bias-reducing bootstrap estimates [27]. The results are confirmed by a graphical method called CUSUNORO plots [28,29]. A local approach for the sensitivity analysis of the MSR was proposed in [30], based on the adjoint technique. To overcome some of the common drawbacks of local approaches, a global approach has been applied, for the first time, to study an MSR system. More specifically, local approaches evaluate the importance of one input parameter at a time (neglecting interactions), with other inputs being kept constant at their nominal value, whereas global approaches consider the interactions among all the input parameters and their variability on their whole support of distribution. Consequently, global approaches are more appropriate when applied to analyze strongly non-linear systems, such as MSRs,

whose model input parameters typically interact with each other. In this frame, the paper aims at filling the gap, by proposing a global sensitivity analysis of a fast-spectrum MSR.

Following the SA, a surrogate model, based on point kinetics equations, is developed, in which only the most relevant parameters are retained. The purpose of the surrogate model is to achieve lower computational times than the high-fidelity models, providing a simulation tool that is suitable for real-time analysis. Regarding the accuracy of the new approach, the two models are compared, simulating two different accidental transients in an MSR, namely, the super-prompt-critical reactivity insertion and the loss of heat sink accidents.

Point kinetic models of the MSR were proposed in [31–35]. Furthermore, simplified models of MSRs were also developed by using other approaches, e.g., 1D system codes [36]. Independently from the specific approach selected to develop the simplified model, this research aims at demonstrating the suitability of global SA to provide a synthetic overview of the most relevant parameters to be used for model surrogation. In other words, the purpose of this work is (i) to identify the most relevant physical parameters affecting MSR behavior using a global SA approach, (ii) to develop a surrogate MSR model based on the reduced set of most relevant parameters identified by the sensitivity analysis, and (iii) demonstrate that this surrogate model is comparable to more complex ones in terms of accuracy but computationally cheaper. Therefore, the novelty of this work as compared to state-of-the-art literature is that it proves global SA as an effective tool in support of the development of simplified, control-oriented models of MSRs, which are suitable for real-time decision making. It is underlined that, even though point kinetics is selected to develop the surrogate model in this work, the proposed technique can be flexibly applied to any other simplified modeling approach, such as 1D system codes.

The remainder of the paper is organized as follows. Section 2 describes the main features of MSR systems. Section 3 discusses the high-fidelity model. Section 4 presents the SA methods in use. Section 5 reports the SA results. Then, in Section 6, the surrogate model is developed, and in Section 7 its results are compared to the high-fidelity model. Finally, conclusions are provided in Section 8.

## 2. MSRs

A schematic representation of a fast-spectrum MSR layout is shown in Figure 1 and the main design parameters are listed in Table 1. The high-fidelity simulations presented in Section 7 are carried out using the MSR system described in Figure 1 and Table 1. The salt mixture, which serves simultaneously as fuel and coolant, is circulated through 16 external circuits, where the power produced by fissions is removed by heat exchangers. The helium bubbles are injected in the cold legs, at the bottom of the system, and they are removed at the top, in the hot leg region.

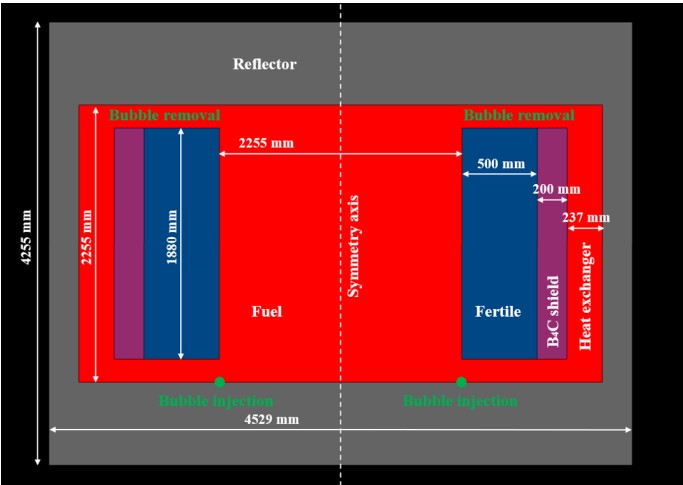

**Figure 1.** Layout of the fast-spectrum MSR studied in this work.

**Table 1.** Main design parameter values assumed in this work [1,16].

| Parameter | Value |
|---|---|
| Nominal power | 3000 MW$_{th}$ |
| Fuel inlet temperature | 923 K |
| Fuel outlet temperature | 1023 K |
| Total salt volume | 18 m$^3$ |
| Fuel composition (mol. %) | LiF (77.5)—ThF$_4$ (20.0)—$^{233}$UF$_4$ (2.5) |
| Injector diameter | 3 mm |

## 3. The Multiphysics High-Fidelity Model

The multiphysics high-fidelity model solves, at each time step, the system thermal-hydraulics and neutronics in two different cycles, as sketched in Figure 2. The thermal-hydraulics sub-solver is based on the standard OpenFOAM [37] solver "*twoPhaseEulerFoam*" for the compressible fluid and the bubble modeling, whereas neutronics is described by multi-group neutron diffusion, SP$_3$ and discrete ordinate transport sub-solvers [19,20].

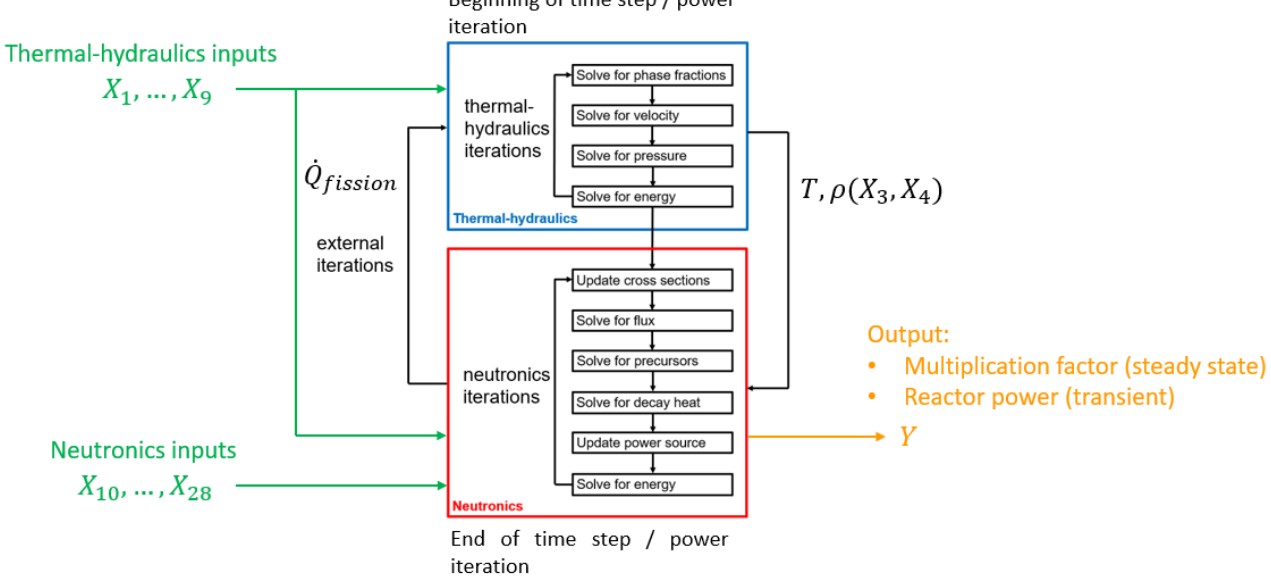

**Figure 2.** Logic structure of the multiphysics solver.

The list of the inputs of the multiphysics model, $\overline{X}(X_1, \ldots, X_{28})$ is provided in Table 2, together with their uncertainty distributions. These uncertainty distributions were determined by various authors in previous works, as cited in the last column of Table 2.

**Table 2.** List of the multiphysics model inputs.

| Input | Parameter | Description | Ref. |
|---|---|---|---|
| $X_1$ | Bubble diameter | A log-normal distribution is adopted in this work as well, as commonly done in literature. With regards to the support of such distribution, the bubble diameter is typically assumed to lay in between 1 and 5 mm with most probable values around 3 mm (i.e., for air and water [38–40]) and, in the case of the Molten Salt Reactor Experiment (MSRE), in between 0.127 and 0.508 mm [41]. For the MSR considered in this work, the diameter of the helium bubbles is taken equal to 3 mm, i.e., the most probable diameter, since it can be determined by the helium injector diameter (of 3 mm, see Table 1), as supported by analyses carried out in [42], where different bubble diameter models are compared. In more detail, [42] points out that bubbles injected with 3 mm diameter remain the same size in the whole reactor, without being significantly affected by bubble coalescence and break-up. In this respect, it is worth mentioning that, up to now, a helium bubbling system has never been designed for a fast-spectrum MSR and, therefore, no evidence is available to support different hypotheses on the actual helium bubbles diameter. **Distribution (\*):** $X_1 = x$ [mm] $x \sim Logn(0.83, 0.48)$ | [38–40] |

**Table 2.** *Cont.*

| Input | Parameter | Description | Ref. |
|---|---|---|---|
| $X_2$ | Surface tension | The surface tension of the salt adopted in this work (77.5% LiF 20.0% ThF$_4$ 2.5% $^{233}$UF$_4$) has not been measured yet. A uniform distribution is assumed, that accounts for the uncertainties of correlations for other fluorides.<br>For 46.5% LiF, 11.5% NaF, 42% KF (FLiNaK):<br>$X_2 = 0.2726 - 1.014 \cdot 10^{-4} T[\text{K}]$<br>$\pm 2\%$ of uncertainty.<br>Valid for the temperature range T = (770–1040) K.<br>For 33% LiF, 67% BeF$_2$ (FLiBe):<br>$X_2 = 0.295778 - 0.12 \cdot 10^{-3} T[\text{K}]$<br>$\pm 3\%$ of uncertainty.<br>Valid for the temperature range T = (773.15–1073.15) K.<br>**Distribution:**<br>$X_2 = x \ [\text{N m}]$<br>$x \sim U(0.15, 0.19)$ | [43] |
| $X_3, X_4$ | Fuel density | $\rho = X_3 - X_4 \cdot T[\text{K}]$ where:<br>$X_3 = 4983$<br>$X_4 = 0.882$<br>$\pm 0.9\%$ of error.<br>**Distribution (\*\*):**<br>$X_3 = 4983 \cdot (1 + 0.01x) \ [\text{kg m}^{-3}]$<br>$x \sim N(0, 0.547)$<br>$X_4 = 0.882 \cdot (1 + 0.01x) \ [\text{kg m}^{-3}\text{K}^{-1}]$<br>$x \sim N(0, 0.547)$ | [44,45] |
| $X_5, X_6$ | Fuel specific heat | $c_p = X_5 - X_6 \cdot T[\text{K}]$ where:<br>$X_5 = -1.111$<br>$X_6 = 0.00278$<br>$\pm 10\%$ of error.<br>**Distribution (\*\*):**<br>$X_5 = -1111 \cdot (1 + 0.01x) \ [\text{J kg}^{-1}\text{K}^{-1}]$<br>$x \sim N(0, 6.08)$<br>$X_6 = 0.00278 \cdot (1 + 0.01x) \ [\text{J kg}^{-1}\text{K}^{-2}]$<br>$x \sim N(0, 6.08)$ | [44,45] |
| $X_7, X_8$ | Fuel thermal conductivity | $k = X_7 - X_8 \cdot T[\text{K}]$ where:<br>$X_7 = 0.928$<br>$X_8 = 8.397 \cdot 10^{-5}$<br>$\pm 5\%$ of experimental error.<br>**Distribution (\*\*):**<br>$X_7 = 0.928 \cdot (1 + 0.01x) \ [\text{W m}^{-1}\text{K}^{-1}]$<br>$x \sim N(0, 3.04)$<br>$X_8 = 8.397 \cdot 10^{-5} \cdot (1 + 0.01x) \ [\text{W m}^{-1}\text{K}^{-2}]$<br>$x \sim N(0, 3.04)$ | [44,45] |
| $X_9$ | Fuel kinematic viscosity | $X_9 = 5.54 \cdot 10^{-8} \exp\left(\frac{3689}{T[\text{K}]}\right)$<br>$\pm 5\%$ of experimental error.<br>**Distribution (\*\*):**<br>$X_9 = 5.54 \cdot 10^{-8} \exp\left(\frac{3689}{T[\text{K}]}\right) \cdot (1 + 0.01x) \ [\text{m}^2\text{s}^{-1}]$<br>$x \sim N(0, 3.04)$ | [44,45] |
| $X_{10}$ to $X_{28}$ | All neutronics parameters | Normally distributed. Mean and standard deviation evaluated with Serpent (\*\*\*) | See box below for details |

(\*) The log-normal parameters μ and σ are determined so that the most probable bubble diameter value is 3 mm and 90% of the samplings fall into the 1–5 mm range. (\*\*) The normal parameters μ and σ are determined so that μ is equal to the nominal value of the parameter and that 90% of the samplings fall within the experimental error. (\*\*\*) The neutronics parameters are: -Cross sections (diffusion coefficient, fission XS times neutrons per fission, fission XS times fission power, absorption each described by a constant reference value and by a temperature coefficient): $X_{10}$ to $X_{17}$; -Delayed neutron fractions (from group 1 to 8): $X_{18}$ to $X_{25}$; -Decay heat power fractions (from group 1 to 3): $X_{26}$ to $X_{28}$. Uncertainties on nuclear data are calculated with the Serpent Monte Carlo code [46], using 100 million neutron histories and obtaining a 5 pcm 1-σ uncertainty on the multiplication factor. The JEFF-3.1.1 nuclear data library [47] is selected, which divides delayed neutron precursors into eight groups. On the other hand, three groups are adopted for decay heat precursors, based on results obtained by Aufiero et al. in [9]. For more detail on the Serpent model adopted in this work, the reader is referred to [19].

The thermal-hydraulics solver, fed with the thermal-hydraulics inputs $X_1$ to $X_9$, finds the phase fractions, the velocity of both phases, pressure, and temperature. Picard iterations are performed until convergence is reached for the solution of the thermal-hydraulic part of the problem. Then, the neutronics solver, fed with the neutronics inputs $X_{10}$ to $X_{28}$, finds the flux, the delayed neutron precursors, and the decay heat. Once the flux (and the fission power in turn) and the decay heat are known, the volumetric power source field is updated and the energy equation is solved again (for this reason, the thermal-hydraulics inputs $X_1$ to $X_9$ are also provided to the neutronics solver). Once the new temperature

and density fields of the fuel are calculated, the cross sections are updated and the cycle is repeated with Picard iterations until convergence is reached. In addition, a certain number of external iterations between the thermal-hydraulics and the neutronics sub-solvers is performed. In more detail, the temperature $T$ and the fuel density $\rho(X_3, X_4)$, calculated by the thermal-hydraulics solver, are passed to the neutronics cycle, whereas the fission power $\dot{Q}_{fission}(X_1, \ldots, X_{28})$, calculated by the neutronics solver, is passed to the thermal-hydraulics cycle. The external iterations are particularly important in fast transients, in which the large thermal expansions due to steep power excursions have a strong impact on the fuel velocity field.

This model can be used in two different modes:

1.  A time-independent, criticality mode, in which the system multiplication factor is evaluated at steady-state conditions. To this aim, a power iteration routine, based on the $k$-eigenvalue method [48] is implemented into the neutronics module. In this case, the main output is represented by the multiplication factor.
2.  A time-dependent mode, for the analysis of operational as well as accidental transients. The main output provided by the transient mode is the reactor thermal power.

In both cases, the temperature and velocity fields of the fuel and of the gas bubbles, the void fraction distribution, the pressure fields, and the precursor density distributions are provided as output.

More details on the thermal-hydraulics and neutron diffusion modules are provided in the following sections.

### 3.1. The Thermal-Hydraulics Model

The need for a two-phase thermal-hydraulics solver is due to the presence of gaseous fission products inside the reactor. To this aim, the "*twoPhaseEulerFoam*" solver available in the OpenFOAM library is used, which implements a Euler–Euler approach [21]. Each phase is treated as a continuum interpenetrating each other and is described with averaged conservation equations. Due to the averaging process, phase fractions are introduced into the governing equations.

The "*twoPhaseEulerFoam*" solver is selected since it is well established in the Open-FOAM community and widely validated for a broad range of applications, spacing from turbulent, multiphase flows to boiling flows, chemical engineering and pharmaceutical applications, and powder technology applications. For a detailed overview, the reader is referred to [42].

Compared to other approaches for bubbly flows (e.g., Lagrangian–Lagrangian and Eulerian–Lagrangian), the Euler–Euler approach is characterized by lower computational requirements, and therefore is suitable for the simulation of high-Reynolds and large-scale systems, which is the case for the MSFR. In fact, considering an average fuel density of $4125 \, \text{kg/m}^3$, an average fuel velocity of about 1.2 m/s, a diameter of 2.26 m, and a dynamic viscosity of $10^{-2}$ Pa s [1], the Reynolds number is in the order of $10^6$, implying a fully turbulent flow regime. For these reasons, the Euler–Euler approach is the preferred method for many practical applications and is adopted also in this work. Furthermore, the adoption of a Euler–Euler approach is compatible with the gas fractions expected in fast-spectrum MSRs implementing helium bubbling for fission gas removal, typically in the order of 1% [18–20].

The mass and momentum conservation equations for the two phases read:

$$\begin{cases} \frac{\partial\left(\rho_j(X_3,X_4)\alpha_j\right)}{\partial t} + \nabla\cdot\left(\rho_j(X_3,X_4)\alpha_j\boldsymbol{u}_j\right) = S_j \quad j = fuel, \; gas \\ \frac{\partial\rho_j(X_3,X_4)\alpha_j\boldsymbol{u}_j}{\partial t} + \nabla\cdot\left(\rho_j(X_3,X_4)\alpha_j\boldsymbol{u}_j\boldsymbol{u}_j\right) = \\ \nabla\cdot\alpha_j\Big[-p\boldsymbol{I} + (\mu(X_3,X_4,X_9) + \mu_t(X_3,X_4,X_9))\left(\nabla\boldsymbol{u} + (\nabla\boldsymbol{u})^T\right) - \frac{2}{3}\mu(X_3,X_4,X_9)(\nabla\cdot\boldsymbol{u})\boldsymbol{I}\Big] \\ \qquad +M_j(X_1,X_3,X_4,X_9) \end{cases} \quad (1)$$

A mass source term $S_j$ is considered in the continuity equation to model gas injection and extraction in the reactor. The term $M_j$ appears in the averaged momentum equations of each phase due to non-linearity, which requires closure equations. This term considers the momentum transfer between the two phases, due to the forces acting at the liquid–gas interface, namely the lift, the drag, virtual mass forces, and turbulent dispersions. Several models are implemented into the solver to describe the inter-phase terms and to close the momentum equation [49,50]. These closure models usually depend on the bubble diameter, which is sampled as described in Table 2.

The energy equations for the two-phases for the "*twoPhaseEulerFoam*" read:

$$
\frac{\partial \rho_j(X_3, X_4)\alpha_j h_j}{\partial t} + \nabla \cdot \left(\rho_j(X_3, X_4)\alpha_j \boldsymbol{u}_j h_j\right) + \frac{\partial \rho_j(X_3, X_4)\alpha_j k_j}{\partial t} + \nabla \cdot \left(\rho_j(X_3, X_4)\alpha_j \boldsymbol{u}_j k_j\right)
$$
$$
= \alpha_j \frac{\partial p}{\partial t} + \frac{\alpha_j}{\rho_j(X_3, X_4)C_{p,j}(X_5, X_6)} \nabla \cdot \left((K(X_7, X_8) + K_t(X_7, X_8))\nabla h_j\right) + L(X_1, X_3, X_4, X_9)\Delta T \tag{2}
$$
$$
+ \rho_j(X_3, X_4)\alpha_j \boldsymbol{g} \cdot \boldsymbol{u}_j + Q_f(X_{14}, X_{15}) + Q_h(X_{16}, X_{17}, X_{26}, X_{27}, X_{28})
$$

where $L$ is an inter-phase heat transfer coefficient resulting from the averaging process and $\Delta T$ is the temperature difference between the two phases. Also in this case, different models are implemented in the solver and can be chosen to describe $L$, closing the energy equation [51]. In addition, the Lahey k-$\varepsilon$ turbulence model [52] was adopted to account for the contribution of the dispersed gaseous phase on eddy viscosity.

### 3.2. The Neutronics Model

The one-speed formulation of the diffusion equation is adopted in this work:

$$
\frac{1}{v}\frac{\partial \varphi}{\partial t} = \nabla \cdot D(X_{10}, X_{11})\nabla \varphi - \Sigma_a(X_{16}, X_{17})\varphi + S_n(1 - \beta(X_{18}, \ldots, X_{25})) + S_d \tag{3}
$$

Note that the neutron velocity $v$ is not sampled, since its relative uncertainty is nearly zero, and therefore negligible compared to those of the cross sections and of the diffusion coefficient. The macroscopic cross sections are evaluated by assuming a logarithmic dependence on temperature and a linear dependence on density and on the void fraction due to the helium bubbles, according to the following relation:

$$
\Sigma = \left[\Sigma^o(X_i) + A(X_j)log\frac{T_{fuel}}{T_{ref}}\right]\frac{\rho_{fuel}(X_3, X_4)}{\rho_{ref}}(1 - \alpha_b)
$$
$$
where\ (X_i, X_j) = (X_{12}, X_{13}),\ (X_{14}, X_{15}),\ (X_{16}, X_{17}) \tag{4}
$$

with $\rho_{ref,fuel} = 4125\ \text{kg/m}^3$. The reference term $\Sigma^o(X_i)$ is a group-constant cross section evaluated by Monte Carlo simulation at reference temperature $T_{ref}$ and density $\rho_{ref}$, while $A(X_j)$ is calculated by logarithmic interpolation of two cross sections values, obtained at $T_{ref}$ and at a different temperature (always by Monte Carlo simulation). The suitability of Equation (4) to account for temperature and density feedback on macroscopic cross sections has been verified in [19,20] and validated in [53].

The diffusion coefficient is evaluated with a similar expression:

$$
D = \left[D^o(X_{10}) + A_D(X_{11})log\frac{T_{fuel}}{T_{ref}}\right]\frac{\rho_{fuel}(X_3, X_4)}{\rho_{ref,fuel}}(1 - \alpha_b) \tag{5}
$$

The source terms represent the fission neutrons, the scattering neutrons, and the delayed neutrons, respectively, and are evaluated as follows:

$$
S_n = \bar{v}\Sigma_{f,j}(X_{12}, X_{13})\varphi \tag{6}
$$

$$
S_d = \sum_k \lambda_k c_k \tag{7}
$$

Due to these explicit terms, an iterative procedure among the several groups is required to achieve convergence for the neutronics description. Albedo boundary conditions are adopted at the top and bottom walls of the reactor (axial reflectors) and at the radial wall (blanket salt), in order to limit the domain of the equation set of neutronics to the fuel salt circuit only [8,54].

The precursor balance equations include the diffusion and the transport term to allow for the fuel motion (neglecting the precursor mass transfer from the liquid to the gas phase):

$$
\begin{aligned}
\frac{\partial \rho_l(X_3, X_4)\alpha_l c_k}{\partial t} \quad & + \nabla \cdot (\rho_l(X_3, X_4)\alpha_l \boldsymbol{u}_l c_k) \\
& = \nabla \cdot \left( \rho_l(X_3, X_4)\alpha_l \left( \frac{\nu(X_9)}{Sc} + \frac{\nu_T(X_9)}{Sc_T} \right) \nabla c_k \right) \\
& + \beta_k(X_k) \sum_i \bar{\nu}\Sigma_{f,i}(X_{12}, X_{13})\varphi_i - \lambda_k \rho_l(X_3, X_4)\alpha_l c_k \\
& \textit{where } X_k = X_{18}, \dots, X_{25}
\end{aligned}
\tag{8}
$$

The turbulent Schmidt number $Sc_T$ is set to 0.85, even if no data are specifically available for the diffusion of species in the MSR salt [8].

In order to properly consider the decay heat during accidental transients, the solver is provided with equations that consider the behavior of the isotopes responsible for the decay heat, subdivided in "decay heat groups" in a manner similar to the precursor groups. Actually, the equations implement the balance for the precursor concentration multiplied by the average energy released by that decay group:

$$
\begin{aligned}
\frac{\partial \rho_l(X_3, X_4)\alpha_l d_m}{\partial t} \quad & + \nabla \cdot (\rho_l(X_3, X_4)\alpha_l \boldsymbol{u}_l d_m) \\
& = \nabla \cdot \left( \rho_l(X_3, X_4)\alpha_l \left( \frac{\nu(X_9)}{Sc} + \frac{\nu_t(X_9)}{Sc_t} \right) \nabla d_m \right) \\
& + \beta_{h,m}(X_r) \sum_i E_f \Sigma_{f,i}(X_{14}, X_{15})\varphi_i - \lambda_{h,l} \rho_l(X_3, X_4)\alpha_l d_m \\
& \textit{where } X_r = X_{26}, \dots, X_{28}
\end{aligned}
\tag{9}
$$

where, again, the decay constants $\lambda_{h,l}$ are fixed and do not need to be sampled. The decay heat is of interest for a more accurate evaluation of power density and of temperature evolution during transients. It is assumed that the decay heat is deposited in situ (i.e., where the corresponding precursors decay) without considering photon transport.

In addition, a power iteration routine, based on the k-eigenvalue method [48], is implemented in the neutronics module of the solver for the calculation of the multiplication factor.

## 4. Global SA Methods

In global SA, the model inputs $\boldsymbol{X} = (X_1, \dots, X_k) \in R^k$ are considered as random variables following certain probabilistic distribution. The uncertainty in the inputs propagates through the model to the output so that the output $Y$ (i.e., the multiplication factor) is also a random variable. To identify the most important input parameters affecting the output $Y$, the degree of statistical dependence between $Y$ and $X_i, i = 1 \dots k$ is of concern. The stronger the statistical dependence, the more important we consider the corresponding parameter. Global sensitivity measures are developed to measure the statistical dependence. For example, the *Pearson correlation coefficient* $\rho_{Y,X_i}$ (or Pearson product-moment correlation coefficient) is defined as [55]:

$$
\rho_{Y,X_i} = \frac{\text{Cov}(Y, X_i)}{\sigma_Y \sigma_{X_i}}
\tag{10}
$$

where $\text{Cov}(Y, X_i)$ denotes the covariance between $Y$ and $X_i$, $\sigma_Y$ and $\sigma_{X_i}$ are the corresponding standard deviations, whereas the moment-independent Borgonovo $\delta$ index,

Kolmogorov–Smirnov (KS) index $\beta_i^{KS}\cdot$, and Kuiper (KU) index $\beta_i^{KU}$ are provided in Equations (11)–(13), respectively [26]:

$$\delta_i = E_{X_i}\left[\int\left|f_Y(y) - f_{Y|X_i=x_i}(y)\right|dy\right] \tag{11}$$

$$\beta_i^{KS} = E_{X_i}\left[\sup_{\mathcal{Y}}\left|F_Y(y) - F_{Y|X_i=x_i}(y)\right|\right] \tag{12}$$

$$\beta_i^{KU} = E_{X_i}\left[\sup_{\mathcal{Y}}\left(F_Y(y) - F_{Y|X_i=x_i}(y)\right) + \sup_{\mathcal{Y}}\left(F_{Y|X_i=x_i}(y) - F_Y(y)\right)\right] \tag{13}$$

where the marginal cumulative distribution function (cdf) and density function (pdf) of $X_i$ are denoted by $F_{X_i}$ and $f_{X_i}$, and the cdf and pdf of the model output by $F_Y$ and $f_Y$, respectively.

To keep the computational burden under control, we use a given-data approach to estimate the above sensitivity indices ($\rho_{Y,X_i}, \delta_i, \beta_i^{KS}, \beta_i^{KU}$) [26]. For quantifying the uncertainty in the estimates, the ensemble-based approach proposed in [25] is adopted. Specifically, we ensemble the rankings of $\rho_{Y,X_i}, \delta_i, \beta_i^{KS}, \beta_i^{KU}$ by considering their sum aggregation $R_{sum}$, i.e., the ranking of the sum aggregation is calculated by first taking the sum of the ranking positions of each sensitivity index, then sorting the input parameters according to corresponding $R_{sum,i}$.

The results are confirmed by a graphical visualization tool called the CUSUNORO plot [28,29]. The CUSUNORO curve for a parameter $X_i$ at its quantile $u \in [0,1]$ is given by:

$$c_i(u) = \frac{u}{\sqrt{V[Y]}}E\left[Y - E[Y]\Big|X_i \leq F_{X_i}^{-1}\right] = \frac{1}{\sqrt{V[Y]}}\int_{-\infty}^{F_{X_i}^{-1}(u)}E[Y - E[Y]|X_i = \mathrm{x}]d\mathrm{x}, \quad i = 1\ldots k \tag{14}$$

The curve $c_i(u)$ shows the average change in the standardized output to the mean when the associated input parameter changes to a given quantile $u$. If the corresponding parameter has a significant effect on the model output, the curve $c_i(u)$ has large dispersion from the zero-horizontal line at any $u$. One can also obtain the monotonicity information of the corresponding $X_i$ from the CUSUNORO plot.

## 5. SA Results

The SA methods introduced in Section 4 were applied to analyze the MSR model, using a Monte Carlo sample of size $n = 50$. We used the given-data principle to estimate the Pearson correlation coefficient, Borgonovo $\delta$, KS and KU indices. The bias-reducing bootstrap estimation scheme was used to obtain the error bands.

Figure 3 shows boxplots of bootstrap estimates of the four global sensitivity measures, with the bootstrap size of 100. All four indices rank the fuel density parameters $X_4$ and $X_3$ as the most important input parameters; the fuel density parameters are significantly more important than the others, as indicated by their higher sensitivity estimates that are non-overlapping with the others. The boxplots also suggest that the remaining inputs, except for the fuel density parameters, have limited effects on the output reactor power, and their ranking is not clear.

Table 3 shows the ranking obtained by the bootstrap mean of $\rho_{Y,X_i}, \delta_i, \beta_i^{KS}, \beta_i^{KU}$, together with the sum ensemble ranking $R_{sum,i}$. Again, the fuel density parameters $X_4$ and $X_3$ are ranked as the first and the second, followed by the neutronics parameter $X_{20}$, the mean bubble diameter $X_1$, and the neutronics parameter $X_{15}$.

The CUSUNORO plot in Figure 4 is obtained using the same dataset, where each curve refers to an input parameter. The magnitudes of the deviations from the zero horizontal line can be used to infer information about the strength of the impact. The CUSUNORO plot

shows a clear agreement with the previous results: the fuel density parameters $X_4$ (blue -o-) and $X_3$ (red -Δ-) are significantly more important than the others.

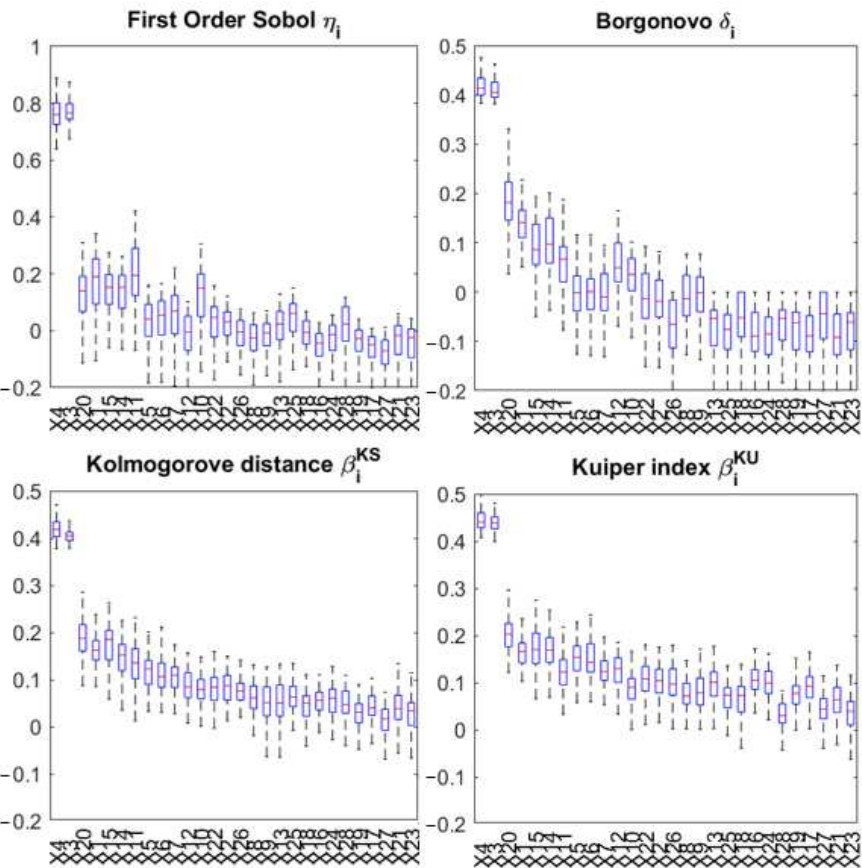

**Figure 3.** Boxplots of bootstrap estimates for global sensitivity measures with $n$ = 50 MSR runs.

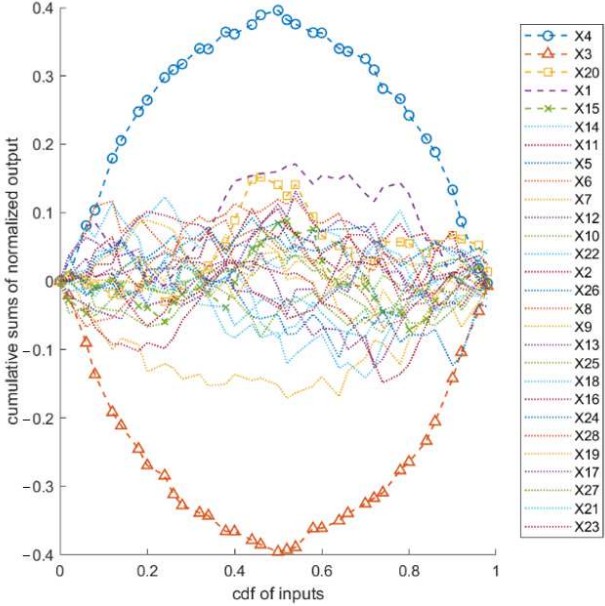

**Figure 4.** The CUSUNORO plot with $n$ = 50 MSR runs.

**Table 3.** Input variable ranking obtained with $n = 50$ MSR runs.

| Sensitivity Measure/Rank | 1st | 2nd | 3rd | 4th |
|---|---|---|---|---|
| Pearson correlation coefficient, $\eta_i$ | 3 | 4 | 11 | 1 |
| Borgonovo δ index, $\delta_i$ | 4 | 3 | 20 | 1 |
| KS index, $\beta_i^{KS}$ | 4 | 3 | 20 | 15 |
| KU index, $\beta_i^{Ku}$ | 4 | 3 | 20 | 15 |
| Ensemble $R_{sum}$ | 4 | 3 | 20 | 1 or 15 |

Furthermore, curves above the horizontal zero line signal a decreasing effect; curves below the horizontal zero line suggest the opposite. From Figure 4, one observes that $X_4$ has a decreasing effect on the output reactor power, whereas $X_3$ shows an increasing effect.

## 6. The Surrogate Model

The high-fidelity multiphysics model presented in Section 3 can produce very accurate solutions, being able to catch phenomena that could not be described with simpler approaches. However, this accuracy comes at the expense of large computational times, which can be as high as about 24 h in three-dimensional full core simulations. Due to this drawback, the high-fidelity model is not adequate for real-time control and decision support. To overcome this issue, a simplified surrogate model is developed in Matlab®, suitable for the real time analysis of the MSR.

As pointed out by the SA results in Section 5, the effect of the density parameters on reactivity are far more important than all the other model parameters. In the light of this, a surrogate model is developed, only considering the density feedback on reactivity.

Point kinetics equations [56] are selected for the estimation of the fission power. In order to simplify the model as much as possible, only one group is considered for the delayed neutron precursors:

$$\frac{d\psi}{dt} = \frac{\widetilde{\rho} - \beta_{eff}}{\Lambda} \psi + \frac{\beta_{eff}}{\Lambda} \eta \tag{15}$$

$$\frac{d\eta}{dt} = \lambda\psi - \lambda\eta \tag{16}$$

where $\psi$ and $\eta$ are the fission power and the precursor density, normalized to their initial values [57]:

$$\psi = \frac{P}{P_{t=0}} \tag{17}$$

$$\eta = \frac{c}{c_{t=0}} \tag{18}$$

The effective delayed neutron fraction is evaluated by [9] and it accounts for the fact that, due to the fuel motion, some of the precursors decay outside of the reactor, where they do not contribute to the fission chain reactor. The other kinetics parameters are obtained by means of Monte Carlo simulation. The density feedback on reactivity is described by means of the following relation:

$$\widetilde{\rho} = \widetilde{\rho}_{ext} + \alpha_\rho (\rho - \rho_{t=0}) \tag{19}$$

where the density feedback coefficient is evaluated by Monte Carlo simulation. The Doppler reactivity feedback and leakage effects are not included on purpose, to demonstrate that the density feedback alone is able to correctly reproduce transients, as suggested by the results of the global sensitivity analysis (which points out that density is the most relevant parameter affecting reactivity).

On the other hand, thermal-hydraulics is described by means of the following equation:

$$mc_p \frac{dT}{dt} = \psi P_{t=0} - \Gamma c_p (T_{out} - T_{in}) \tag{20}$$

where $T$, $T_{in}$, and $T_{out}$ are the average, the inlet, and the outlet fuel temperatures, respectively. Assuming for simplicity that:

$$T = \frac{T_{in} + T_{out}}{2} \tag{21}$$

Equation (20) can be rewritten as follows:

$$\frac{d(\rho V c_p T)}{dt} = \psi P_{t=0} - 2\rho \Gamma c_p (T - T_{in}) \tag{22}$$

All the parameters appearing in Equations (15) to (22) are listed in Table 4. The effective delayed neutron fraction $\beta_{eff}$ is calculated in [8] and accounts for precursor transport within the liquid fuel, while the other neutronics parameters are calculated by Monte Carlo simulation. The thermophysical properties and flow parameters can be found in [1,16]. Note that none of these parameters are among the 28 $X_i$ listed in Table 2. Their nominal values are used to define the initial conditions of the system, whereas transients are evaluated considering only the density variations, which are described by the following equation:

$$\rho = X_3 + X_4 \, T[\text{K}] = 4983 - 0.882 \, T[\text{K}] \tag{23}$$

**Table 4.** Input parameter values adopted in the surrogate model [1,8,16].

| Parameter | Symbol | Value | Unit |
|---|---|---|---|
| Effective delayed neutron fraction | $\beta_{eff}$ | 146 | Pcm |
| Precursor decay constant | $\lambda$ | 0.317 | $s^{-1}$ |
| Mean neutron generation time | $\Lambda$ | 1.147 | $\mu s$ |
| Density reactivity coefficient | $\alpha_\rho$ | 7.85 | $\text{pcm m}^3 \text{ kg}^{-1}$ |
| Fuel inlet temperature | $T_{in}$ | 923 (steady state) | K |
| Reactor volume | $V$ | 9 | $m^3$ |
| Fuel specific heat (at $T$ = 900 K) | $c_p$ | 1391 | $\text{J kg}^{-1} \text{ K}^{-1}$ |
| Initial power | $P_{t=0}$ | 3000 | MW |
| Nominal volumetric flow rate | $\Gamma$ | 4.5 | $\text{m}^3 \text{ s}^{-1}$ |

Therefore, the density coefficients $X_3$ and $X_4$ are the only two parameters retained from the high-fidelity model. Thanks to these parameters, density variations throughout the transient can be evaluated, which in turn are used to evaluate reactivity by means of Equation (17).

## 7. Comparison between the Surrogate and High-Fidelity Models

In this section, the surrogate model developed in Section 6 is tested against the high-fidelity multiphysics model. To this aim, two case studies are analyzed: a super-prompt-critical 500 pcm reactivity insertion, starting from zero power, and a loss-of-heat-sink accident. Due to strong nonlinearity, these transients constitute a tough test set to verify the surrogate model. Additionally, the selected case studies are of great interest from an engineering point of view, representing accidental scenarios that could take place in MSRs. Please note that the high-fidelity simulations are carried out using the geometry presented in Figure 1.

### 7.1. Super-Prompt-Critical Reactivity Insertion

In this section, the accidental super-prompt-critical reactivity insertion is studied. The initial fuel temperature is equal to 900 K. Zero power is approximated with an initial power of 3 MW; in addition, zero void fraction is assumed during the transient. It is also assumed that the heat exchanger secondary temperature is equal to $T_{in} = 923$ K and that perfect heat transfer takes places between the primary and secondary loops. In this way, $T_{in}$ does not change with time. This simplifying assumption is made in both the high-fidelity and surrogate calculations, so that results are coherent each with the other. This assumption allows for simpler modeling of the heat exchanger, avoiding accounting for the secondary loop. At the same time, this simplification is not expected to significantly affect the accuracy of results, since the transient characteristic time (few milliseconds) is much lower than the fuel recirculation time (around 4 s [45]).

In Figure 5, the power excursions resulting from the reactivity insertion evaluated by the multiphysics solver is plotted with a blue line, whereas the surrogate model results are in green. Compared to the multiphysics model, the surrogate model correctly describes the system dynamics, well predicting both the height as well as the shape of the power excursion. In more detail, the peak power predicted by the two curves only differs by 8%. The surrogate model curve shows a slight delay (~0.7 ms) with respect to the high-fidelity model. It is underlined that the macroscopic effect of the transient on the reactor is more determined by the peak power, rather than by the time delay between the two curves. In this regard, the surrogate model can accurately predict the peak power, thus, constituting a useful tool for the analysis of these reactivity-driven accidental transients.

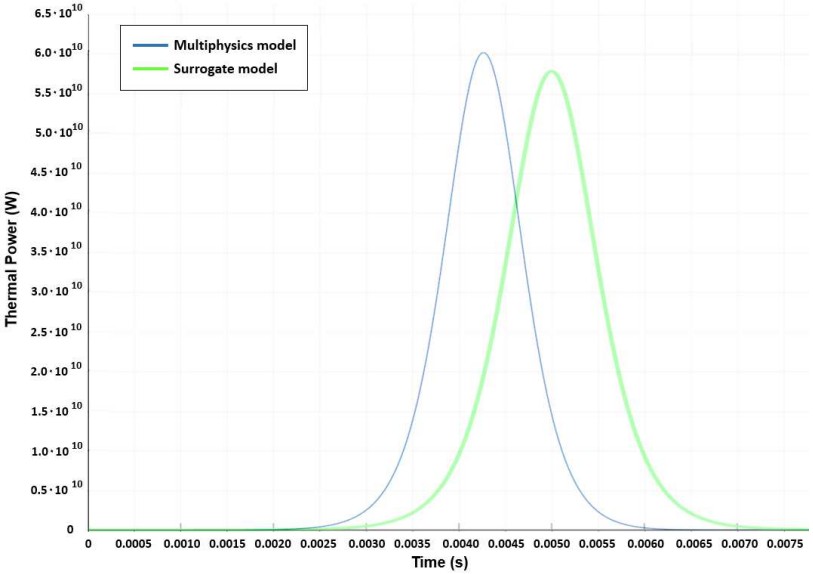

**Figure 5.** Power transient predicted by the multiphysics (blue curve) and by the surrogate model (green curve).

Concerning runtimes, the high-fidelity simulation is carried out in about 10 h on a cluster, using a $2 \times 24$-core Intel Xeon 8160 CPU, whereas the surrogate model requires less than one second on a laptop with an Intel i7-6700HQ CPU. Therefore, the computational burden reduction is clear, at the expense of slightly less accurate results.

### 7.2. Loss of Heat Sink

In this section the loss-of-heat-sink accident is studied. The accident is modeled as a sudden drop to zero of the heat transfer coefficient in the heat exchanger. As the initial condition, the reactor is considered at nominal power (3000 MW) and operating temperature (973 K). The time evolution of the core power following the accident is plotted

in Figure 6. Due to the absence of cooling, the inlet temperature $T_{in}$ is time dependent and always equal to the outlet (and average) temperature.

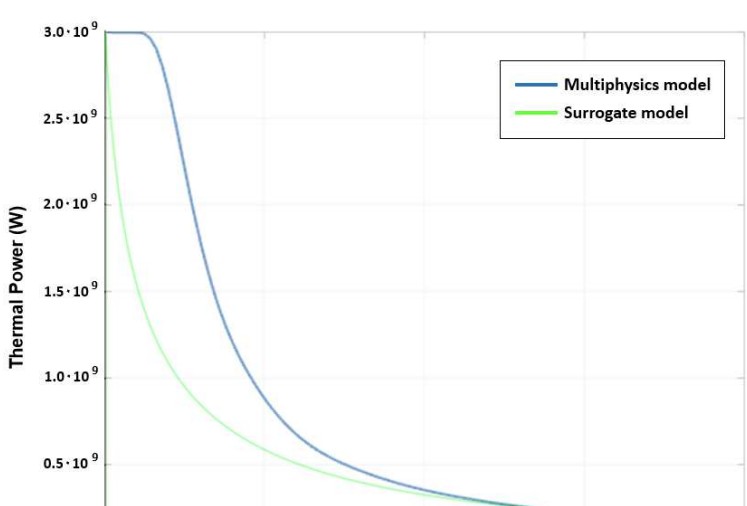

**Figure 6.** Power transient predicted for the loss-of-heat-sink accident by the multiphysics (blue curve) and the surrogate models (green curve).

The multiphysics and surrogate model predictions are in good agreement, only differing by a delay (~1 s) in the first part of the transient. This is because in the high-fidelity model the fuel takes a finite time to circulate through the primary loop, thus, delaying the density feedback on reactivity. Indeed, the recirculation time is 4 s [45], and the fuel takes about 1 s to move from the heat exchanger to the reactor cold leg. This effect cannot be observed when simulating the accident using the surrogate model (since the model is zero-dimensional), thus, explaining the 1 s delay between the two simulations. This small discrepancy, however, is not expected to have a macroscopic effect on the reactor during the accidental transient.

The high-fidelity simulation takes about 2 h on a cluster, using a $2 \times 18$-core Intel Xeon 8160 CPU, whereas the surrogate model requires less than one second on a laptop with an Intel i7-6700HQ CPU. Again, the adoption of the surrogate model allows for a strong reduction of runtime while preserving accuracy.

## 8. Conclusions

The analysis of MSRs is typically carried out by means of complex multiphysics tools, coupling neutronics, thermal-hydraulics, and precursor transport in the same simulation environments. These high-fidelity models can provide a very accurate solution, since they are able to describe physical phenomena that could not be caught by simpler approaches. However, their high computational requirements hinder their application for real-time control and decision support.

To overcome this issue, an SA was carried out on a multiphysics model coupling multi-group neutron diffusion equations with a two-phase, compressible thermal-hydraulics solver for MSR reactivity control. The SA results showed that, among all the model parameters, density is by far the one affecting the reactivity the most. In light of this, a surrogate model was developed, based on point kinetics equations, in which the reactivity feedback depends solely on the fuel density.

The surrogate model was, then, tested against the high-fidelity one, using the simulations of two different accidental transients in an MSR, namely, the super-prompt-critical reactivity insertion and the loss of heat sink accidents. The results point out that the two approaches yield very similar results, in terms of accuracy. However, the surrogate model

runtimes are four orders of magnitude lower, compared to the high-fidelity ones, making it suitable for real time analysis. In particular, the surrogate model can be useful for control purposes, allowing for fast estimations of fission power variations resulting from reactivity insertions or changes in important design parameters such as the fuel temperature or the fuel flow rate. Even though point kinetics were selected to develop the surrogate model, it is worth noting that the proposed technique can be applied to any other simplified modeling approach (e.g., to a 1D system code).

The results of this research point out that global sensitivity analysis approaches are an effective tool to support the development of simplified models of MSR systems, thanks to their ability to identify the most relevant physical parameters. Due to their simplicity, these models can be employed for control purposes, unlike more complex high-fidelity models whose computational requirements are too high to be used for real-time decision-making.

**Author Contributions:** Conceptualization, E.C., X.L., A.C., F.D.M. and E.Z.; methodology, E.C., X.L., A.C., F.D.M. and E.Z.; software, E.C., X.L. and F.D.M.; investigation, E.C., X.L. and F.D.M.; resources, A.C., F.D.M. and E.Z.; data curation, E.C., X.L. and F.D.M.; writing—original draft preparation, E.C., X.L. and F.D.M.; writing—review and editing, E.C., X.L., A.C., F.D.M. and E.Z.; supervision, A.C., F.D.M. and E.Z. All authors have read and agreed to the published version of the manuscript.

**Funding:** This research received no external funding.

**Data Availability Statement:** Research data will be made available by the authors upon request.

**Conflicts of Interest:** The authors declare no conflict of interest.

## Nomenclature

**Latin symbols**

| | |
|---|---|
| $c$ | Delayed neutron precursor density, $\text{m}^{-3}$ |
| $D$ | Neutron diffusion coefficient, m |
| $d$ | Decay heat precursor density, $\text{W m}^{-3}$ |
| $h$ | Specific enthalpy, $\text{J kg}^{-1}$ |
| $K$ | Modified thermal diffusivity, $\text{J m}^{-1} \text{ s}^{-1} \text{ K}^{-1}$ |
| $k$ | Specific kinetic energy, $\text{J kg}^{-1}$ |
| $k_{eff}$ | Effective multiplication factor, $-$ |
| $L$ | Inter-phase heat transfer coefficient, $\text{W m}^{-3} \text{ K}$ |
| $M$ | Inter-phase momentum transfer, $\text{kg m}^{-2} \text{ s}^{-2}$ |
| $p$ | Pressure, Pa |
| $pcm$ | per cent mille ($=10^5$) |
| $Q$ | Power source density, $\text{W m}^{-3}$ |
| $S$ | Mass source, $\text{kg m}^{-3} \text{ s}^{-1}$ |
| $t$ | Time, s |
| $\boldsymbol{u}$ | Velocity, $\text{m s}^{-1}$ |
| $v$ | Neutron velocity, $\text{m s}^{-1}$ |

**Greek symbols**

| | |
|---|---|
| $\alpha$ | Gas fraction, $-$ |
| $\beta$ | Delayed neutron precursor fraction, $-$ |
| $\beta_h$ | Decay heat energy fraction, $-$ |
| $\Delta T$ | Inter-phase temperature difference, K |
| $\lambda$ | Delayed neutron precursor decay constant, $\text{s}^{-1}$ |
| $\lambda_h$ | Decay heat precursor decay constant, $\text{s}^{-1}$ |
| $\mu$ | Dynamic viscosity, Pa s |
| $\nu$ | Kinematic viscosity, $\text{m}^2 \text{ s}^{-1}$ |
| $\bar{\nu}$ | Mean neutrons per fission, $-$ |
| $\rho$ | Density, $\text{kg m}^{-3}$ |
| $\Sigma$ | Macroscopic cross section, $\text{m}^{-2}$ |
| $\varphi$ | Neutron flux (diffusion equation), $\text{m}^{-2} \text{ s}^{-1}$ |

**Subscripts–superscripts**

| | |
|---|---|
| $a$ | Absorption |
| $b$ | Bubble |
| $d$ | Delayed |
| $f$ | Fission |
| $h$ | Decay heat |
| $i$ | Neutron energy group |
| $j$ | Phase |
| $k$ | Delayed neutron precursor group |
| $l$ | Liquid |
| $m$ | Decay heat precursor group |

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
