# Peer review of "Sensitivity-Analysis-Driven Surrogate Model for Molten Salt Reactors Control"

_jne, doi:10.3390/jne3040016_

Round 1
Reviewer 1 Report (Previous Reviewer 3)
I feel my previous comments where properly addressed. I would only recommend uploading the simplified model to an open source github and linking to it in the paper, but this is not necessary since the model is simple enough.
Author Response
Reviewer 1:
I feel my previous comments where properly addressed. I would only recommend uploading the simplified model to an open source github and linking to it in the paper, but this is not necessary since the model is simple enough.
Response:
Thank you for having appreciated the work and the suggestions provided. Research material will be made available by the authors upon request. We will also consider the uploading an open source model on github.
Reviewer 2 Report (Previous Reviewer 1)
Most comments are well responsed by the revised manuscript. But it is incorrect for the response that "ORNL-TM-3464 refers to fission gas (Xe) bubbles ". In ORNL-TM-3464, helium bubbling system was simulated to remove the fission products. The diameter of helium bubble was suggested to be 0.005~0.02 inch (0.127mm~0.508mm). It is not the diameter of fission gas (Xe) bubbles. Please carefully check ORNL-TM-3464. In this case, please specify the reasons why the authors adopt the diameter of helium bubble by referring to the study of air in water, instead of that of helium bubbles suggested by ORNL-TM-3464.
Author Response
Reviewer 2:
Most comments are well responsed by the revised manuscript. But it is incorrect for the response that "ORNL-TM-3464 refers to fission gas (Xe) bubbles ". In ORNL-TM-3464, helium bubbling system was simulated to remove the fission products. The diameter of helium bubble was suggested to be 0.005~0.02 inch (0.127mm~0.508mm). It is not the diameter of fission gas (Xe) bubbles. Please carefully check ORNL-TM-3464. In this case, please specify the reasons why the authors adopt the diameter of helium bubble by referring to the study of air in water, instead of that of helium bubbles suggested by ORNL-TM-3464.
Response:
Thank you for raising this point. Considering that
(1) we are assuming a bubble injector of 3 mm
(2) in reference [41] it is claimed that, for the same fast MSR layout, the bubble diameter is mainly determined by the injector diameter
in this work, we assume that the diameter of helium bubbles to be 3 mm, instead of the values provided in ORNL-TM-3464, supported by the outcomes of the studies presented in [38, 39, 40], in which the bubbles diameter distribution for bubbles is of comparable size.
To clarify these premises, the manuscript has been changed as follows:
A log-normal distribution is adopted in this work as well, as commonly done in literature. With regards to the support of such distribution, the bubble diameter is typically assumed to lay in between 1 and 5 mm with most probable values around 3 mm (i.e., for air and water [38, 39, 40]) and, in the case of the Molten Salt Reactor Experiment (MSRE), in between 0.127 and 0.508 mm [41].
For the MSR considered in this work, the diameter of the helium bubbles is taken equal to 3 mm, i.e. the most probable diameter, since it can be determined by the helium injector diameter (of 3 mm, see Table 1), as supported by analyses carried out in [42], where different bubble diameter models are compared. In more detail, [42] points out that bubbles injected with 3 mm diameter remain the same size in the whole reactor, without being significantly affected by bubble coalescence and break-up. In this respect, it is worth mentioning that, up to now, a helium bubbling system has never been designed for a fast-spectrum MSR and, therefore, no evidences are available to support different hypotheses on the actual helium bubbles diameter.

This manuscript is a resubmission of an earlier submission. The following is a list of the peer review reports and author responses from that submission.
Round 1
Reviewer 1 Report
Overall, the innovation of the study is not sound. Actually, parameter sensitivity analysis for the MSR has been systematically studied (e.g., Jeong et al., 2020. Adjoint-based sensitivity analysis of circulating liquid fuel system for the multi-physics model of molten salt reactor. Int J Energy Res. 2020;1–20.). Meanwhile, the surrogate model proposed by this study is a point kinetic model, which has also been extensively studied for MSR with a high validity (Nuclear Engineering and Design, 2009, 239 (12): 2778-2785; Progress in Nuclear Energy, 2009, 51: 624-636; Annals of Nuclear Energy, 53, 309-319; Journal of nuclear science and technology, 45(6), 575-581; NUCLEAR SCIENCE AND TECHNIQUES, 2014, 25, 030602, et al.). The gap between this study and previous studies is not provided. In addition, the analysis model is not clear, making the results presented in this study hard to be understood. From the literatures cited in this study, I guess that the analysis model is MSFR, but the reactor system presented in Figure 1 is a thermal MSR moderated by the graphite, namely, the description for manuscript is very confusing. Furthermore, some equivalent treatments have been made in the multi-physics calculation model, e.g., cross section process in terms to density, without validation, leading to the doubtful calculation results. There are also other problems in the manuscript,
1. The diameter of helium bubble refers to the study of air in the water, which is significantly different from that the helium bubble in the molten salt. Actually, many efforts have be made for helium bubbling study of MSR (Engel, J. R., et al. Xenon behavior in the molten salt reactor experiment. Oak Ridge National Lab.. 1971, ORNL-TM--3464.), and the diameter of helium bubble in the molten salt was suggested to be 0.005~0.02 inch.
2. It is unclear how to obtain the macroscopic cross sections. Was the resonance effect considered?
3. The variation range of input parameters is set without providing enough reasons.
4. Literature citation is improper, e.g., the neutronics parameters in Table 2 refers to reference 38, whilst it is just a study of code introduction without providing specific cross section information.
5. Typos: “Section Error! Reference source not found”; “In table 2, typos: ‘For 33% 2LiF, 67% BeF2 (FLiBe):”, etc.
From the above, I would suggest to return this manuscript and seek further publication in the future with a clear address on the concerns mentioned above.
Reviewer 2 Report
The paper is very interesting in the most comprehensive multi-physical coupling and sensitivity analysis ability. Some questions are below:
1. Figure 1 may be misused, is it a fast molten salt reactor? mark out the bubble system and the hx.
2. In table 1, why 8 groups delayed neutron fractions are used rather than 6 groups? The same question for the decay heat.
3. How is the bubble diameter used in the two–phase flow calculation? For the momentum transfer calculation? Is it reasonable to treat the bubbles as a continuum interpenetrating fluid? Since the volume fraction of bubbles is very low (<1‰, as observed in MSRE), and the bubble size is very relevant with the velocity of fluid.
4. What are decay heat groups used for? For precise power distribution? Does photon transportation have been considered? Or treat the decay heat as in situ deposits?
5. Line 233, typo error.
6. In section 5, what’s the input of the 50 MSR sample? It’s welcome to show the input δXs. what’s the output Y?
7. Is it reasonable to use the one group point kinetics equations even the effective delayed neutron fraction is evaluated? Think about that in a case with fall power, the returned delayed neutron will has a bigger reactivity insert.
8. Have you considered the other reactivity factors, such as dopple effect, leakage?
9. Is the Tin unchanged with time in equation (22)?
10. The title of section 6 is same to that of section 7.
11. The results in Fig.5 are similar, which may be the reason of the used simple physical model.
12. The results in Fig. 6 seem not be consistent.
13. The transient analysis of MSRs can be well simulated by one-dimensinal system code, like relap, with less computation time since the molten salt is a single phase flow in most cases. I don’t think the surrogate model used in this paper is reasonable and necessary.
Reviewer 3 Report
There are some small editing that needs to be done (e.g. Monten Salt Reactors in abstract, Section 6 and 7 have the same name, etc.).
There needs to be more description of the MSR design being modeled in Section 2. Based off the reactor description in the body and table, I believe system being model is based off of the MSFR, a fast spectrum MSR. Most of the refernces are on the MSFR. However, the system in Figure 1 appears to be a thermal MSR based off of the single fluid breeder reactor by ORNL.
Table 2, the “All Neutronic Parameters” has Serpent cited for the values as the source. Does that mean these that these parameters are calculated using Serpent? If so, it is more important to cite how the values were calculated in Serpent, with either a link to model or a paper that describes the Serpent model.
Can you give some detail on how βeff is calculated. Was it volumetric flow or was this computed in thermal hydraulic model?
In Table 4, The mean neutron generation time is 1.147 Ms (Capital M usually denotes Mega) which is impossible, I think this is a typo and is meant to say 1.147 ms. Given this mean neutron generation time My guess is the reactor modeled is a thermal system, but the reference for the table is for fast reactors. A fast reactor would likely have a mean neutron generation time somewhere within an order of magnitude of a mircosecond. If that could be cleared up and also referenced in the previous table.
Additionally with Table 4, initial power is set to 3 MW, but Table 1 and Figure 6 have power at 3 GW.
I the loss of heat sink surrogate model results, the authors mention that the time delay is due to the salt already in transit from the heat exchanger to the core. Could the surrogate model not just start calculating one second in? Or maybe turn the inlet temperature into a time series that changes after one second?